# On Möbius gyrogroup and Möbius gyrovector space

**Kurosh Mavaddat Nezhaad⋆ and Ali Reza Ashrafi**

Department of Pure Mathematics, Faculty of Mathematical Sciences,
University of Kashan, Kashan 87317-53153, I. R. Iran

⋆ kuroshmavaddat@gmail.com

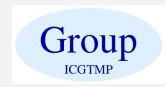
## Abstract

Gyrogroups are new algebraic structures that appeared in 1988 in the study of Einstein's velocity addition in the special relativity theory. These new algebraic structures were studied intensively by Abraham Ungar. The first gyrogroup that was considered into account is the unit ball of Euclidean space $\mathbb{R}^3$ endowed with Einstein's velocity addition. The second geometric example of a gyrogroup is the complex unit disk $\mathbb{D} = \{z \in \mathbb{C} : |z| < 1\}$. To construct a gyrogroup structure on $\mathbb{D}$, we choose two elements $z_1, z_2 \in \mathbb{D}$ and define the Möbius addition by $z_1 \oplus z_2 = \frac{z_1 + z_2}{1 + \bar{z}_1 z_2}$. Then $(\mathbb{D}, \oplus)$ is a gyrocommutative gyrogroup. If we define $r \odot x = \frac{(1+|x|)^r - (1-|x|)^r}{(1+|x|)^r + (1-|x|)^r} \frac{x}{|x|}$, where $x \in \mathbb{D}$ and $r \in \mathbb{R}$, then $(\mathbb{D}, \oplus, \odot)$ will be a real gyrovector space. This paper aims to survey the main properties of these Möbius gyrogroup and Möbius gyrovector space.

## 1 Introduction

Throughout this paper, $\hat{\mathbb{C}} = \mathbb{C} \cup \infty$ denotes the extended complex plane and $\mathbb{C}$, $\mathbb{D} = \{z \in \mathbb{C} : |z| < 1\}$ is the complex open unit disk. We refer the interested readers to consult [1,2] for more information on this topic.

If $G$ is a non-empty set and "+" is a binary operation, then $(G,+)$ is called a groupoid. A permutation $f : G \to G$ with this property that $f(x+y) = f(x) + f(y)$, $x, y \in G$, is said to be an automorphism of $G$. The set of all automorphisms of $G$ is denoted by $\text{Aut}(G)$. A gyrogroup [3] is a groupoid $(G,+)$ satisfying the following axioms:

1. $G$ has a left identity under the binary operation, "+".

2. Each element of $G$ has a left inverse.

3. There exists a mapping gyr : $G \times G \to \text{Aut}(G)$ which satisfies the following conditions:

   (a) For each three elements, $a$, $b$, and $c$ of $G$, $a + (b + c) = (a + b) + \text{gyr}[a, b]c$. This property is named the left gyroassociativity of $G$.

   (b) For every two elements $a, b \in G$, $\text{gyr}[a + b, a] = \text{gyr}[a, b]$. This equality is called the left loop property.

It is custom to wite $\text{gyr}[a, b]$ as $\text{gyr}(a, b)$. This algebraic structure share interesting analogous theorems with classical group theory.

If $(G, \oplus)$ is a gyrogroup, then the following properties of $G$ are important in the context of gyrogroup theory [4]:

1. If $a \oplus b = a \oplus c$ then $b = c$; (general left cancelation law)

2. For each $a \in G$, $\text{gyr}[0, a] = I$;

3. If $x$ is a left inverse of $a$, then $\text{gyr}[x, a] = I$;

4. $\text{gyr}[a, a] = I$;

5. Every left identity is a right identity;

6. The left identity is unique;

7. Every left inverse is a right inverse;

8. The left inverse of each element is unique;

9. For arbitrary elements $a, b \in G$, $\ominus a \oplus (a \oplus b) = b$; (left cancelation law)

10. For arbitrary elements $a, b, x \in G$, $\text{gyr}[a, b]x = \ominus(a \oplus b) \oplus (a \oplus (b \oplus x))$; (gyrator identity)

11. For all elements $a, b \in G$, $\text{gyr}[a, 0] = \text{gyr}[0, b] = I$ and $\text{gyr}[a, b]0 = 0$;

12. For arbitrary elements $a, b, x \in G$, $\text{gyr}[a, b]\ominus x = \ominus \text{gyr}[a, b]x$.

The Möbius transformations are defined as the linear fractional transformations $f(z) = \frac{az+b}{cz+d}$ of $\hat{\mathbb{C}}$, where $a, b, c$ and $d$ are complex numbers satisfying $ad - bc \neq 0$. These transformations constitute a group under composition of functions. This group is isomorphic to the group $PGL(2, \mathbb{C}) = \frac{GL(2,\mathbb{C})}{Z(GL(2,\mathbb{C}))}$. By restricting conditions to $2 \times 2$ matrices with $\det A = 1$, we will have the special linear group $\mathbf{SL}(2, \mathbb{R})$, which is well-known that it maps upper-half plane to itself [2].

The Möbius transformations of complex open disk $\mathbb{D}$ are defined as the mappings given by $z \mapsto e^{i\theta} \frac{a+z}{1+\bar{a}z} = e^{i\theta}(a \oplus z)$, where $a, z \in \mathbb{D}$ and $\theta \in \mathbb{R}$. By seeing this transformation as an addition $\oplus$, we will have Möbius addition of complex open unit disk which is given by $x \oplus y = \frac{x+y}{1+\bar{x}y}$, where $x, y \in \mathbb{D}$, and $\bar{x}$ denotes the conjugate of $x$. It is easy to see that Möbius addition is neither associative nor commutative. For the sake of fixing the lack of associativity and commutativity, Abraham Ungar introduced the gyrations of these algebraic structures as

$$\text{gyr}[x, y]z = \frac{x \oplus y}{y \oplus x}z = \frac{1 + x\bar{y}}{1 + \bar{x}y}z \, ,$$

where $x, y, z \in \mathbb{D}$ [5,6]. This definition will make a gyrogroup structure on $\mathbb{D}$. It is well-known that the gyrators of this gyrogroup is not closed under composition of functions. We refer to [7] for applications of gyrogroup theory in non-Euclidean geometry.

## 2 Möbius structures

Recall that the Möbius addition of two elements $x$ and $y$ in $\mathbb{D}$ is defined as $x \oplus y = \frac{x+y}{1+\bar{x}y}$. It is easy to see that for each element $z \in \mathbb{D}$, $\ominus z = -z$. The gyrator $\text{gyr}[a,b]$ is defined as $\text{gyr}[a,b](x) = \frac{a \oplus b}{b \oplus a}x = \frac{1+a\bar{b}}{1+\bar{a}b}x$. This proves that $\text{gyr}[a,b](c \oplus d) = \text{gyr}[a,b](c) \oplus \text{gyr}[a,b](d)$. It is well-known that in each gyrogroup $\text{gyr}^{-1}[a,b] = \text{gyr}[b,a]$ and by definition of a gyrogroup, all gyrations are automorphisms. By definition of a gyrator, $\text{gyr}[a,b](b \oplus a) = a \oplus b$ and so $(\mathbb{D}, \oplus)$ is gyrocommutative.

Following Ungar [6], by some calculations one can see that $\text{gyr}[u,v](w) = w + 2\frac{Au+Bv}{D}$, where $A = -u.w \parallel v \parallel^2 + v.w + 2(u.v)(v.w)$, $B = -v.w \parallel u \parallel^2 - u.w$, $D = 1 + 2u.v + \parallel u \parallel^2 \parallel v \parallel^2$ and $u, v, w \in \mathbb{V}_s$, where $\mathbb{V}_s$ is generalization of Möbius disk gyrogroup, $(\mathbb{D}, \oplus)$ to $s$-ball of $\mathbb{V}$, $\mathbb{V}_s = \{\mathbb{V}_s \in \mathbb{V} : \parallel v \parallel < s\}$, and its inner product and norm, . and $\parallel . \parallel$, are inherited from its space $\mathbb{V}$ and $+$ denotes the addition of vectors in $\mathbb{V}$. Note that the Möbius addition and Möbius scalar multiplication in $\mathbb{V}_s$ reduce to the vector addition and scalar multiplication, respectively, as $s$ tends to infinity.

Ferreira and Ren [8], studied the algebraic structure of Möbius gyrogroups by a Clifford algebra approach. They started from an arbitrary real inner product space of dimension $n$ and then construct a paravector space from which it is possible to study the Möbius gyrogroups. The most important result of Ferreira and Ren is giving a characterization of the Möbius sub-gyrogroups of $\mathbb{D}$.

A triple $(P, \oplus, \odot)$ consisting a non-empty set $P$ together with two binary operations $\oplus$ and $\odot$ are called a real gyrovector space, if for all real numbers $r$, $r_1$, $r_2$ and all elements $x, y, z \in P$ the following are satisfied: $(i)$ $(G, \oplus)$ is a gyrogroup; $(ii)$ $(r_1 + r_2) \odot x = r_1 \odot x + r_2 \odot x$; $(iii)$ $r_1 r_2 \odot x = r_1 \odot (r_2 \odot x)$; $1 \odot x = x$; $\text{gyr}[x,y](r \odot z) = r \odot \text{gyr}[x,y](z)$; $\text{gyr}[r_1 \odot x, r_2 \odot x] = I$. The gyrovector space is the main object of Ungar's theory of analytic hyperbolic geometry [4]. Kinyon and Ungar [9], applied the relationshops between the geometric and algebraic properties of the Möbius gyrovector space to obtain an interesting geometric picture of these objects.

Suppose $\mathbb{I} = (-1, 1)$, $x, y \in \mathbb{I}$ and $r \in \mathbb{R}$. Define $x \oplus y = \frac{x+y}{1+xy}$ and $r \odot x = \frac{(1+x)^r - (1-x)^r}{(1+x)^r + (1-x)^r}$. Then it is easy to see that $\mathbb{I}$ is a real vector space. Kinyon and Ungar [9] presented an interesting discussion to show that this real verctor space can be generalized to the real gyrovector space $(\mathbb{D}, \oplus, \odot)$ in which $x \oplus y = \frac{x+y}{1+\bar{x}y}$ and $r \odot x = \frac{(1+|x|)^r - (1-|x|)^r}{(1+|x|)^r + (1-|x|)^r}\frac{x}{|x|}$, $x, y \in \mathbb{D}$ and $r \in \mathbb{R}$. The first example, $\mathbb{I}$, is an interesting algebraic example for the Euclidean geometry, but the second one, $(\mathbb{D}, \oplus)$, is an important algebraic example for hyperbolic geometry.

Watanabe [10] introduced the notion of gyrolinear indipendence of vectors in $\mathbb{D}$. To define, we assume that a finite subset $A = \{a_1, a_2, \ldots, a_n\}$ of vectors in $\mathbb{D}$ is given. If for each permutation $\sigma \in S_n$, $r_{\sigma(1)} \odot a_{\sigma(1)} \oplus r_{\sigma(2)} \odot a_{\sigma(2)} \oplus \ldots r_{\sigma(n)} \odot a_{\sigma(n)} = 0$ implies that $r_1 = r_2 = \ldots = r_n = 0$, then $A$ is called a gyrolinear independent set of $\mathbb{D}$. By [10, Lemma 8], every vector of a finite gyrolinearly independent set $A$ is non-zero, and every subset of $A$ is also gyrolinearly independent.

Demirel [11] gave an important sharp inequality between members of $\mathbb{D}$. He proved that if $x_1, x_2, \ldots, x_n$ are non-zero elements of $\mathbb{D}$, then $| \oplus_{j=1}^{n} x_j | \leq \oplus_{j=1}^{n} |x_j|$, where $\oplus_{j=1}^{n} |x_j| = |x_1| + |x_2| + \ldots |x_n|$, $|x_j| = |x_j \ominus 0|$, and

$$\oplus_{j=1}^{n} x_j = (\ldots ((x_1 \oplus x_2) \oplus x_3) \oplus \ldots \oplus x_{n-1}) \oplus x_n.$$

Abe and Watanabe [12] proved that every finitely generated gyrovector subspace in the Möbius gyrovector space is the intersection of the vector subspace generated by the same generators and the Möbius ball. They applied this result to present a notion of orthogonal gyrodecomposition and determined its relationship with the orthogonal decomposition. The

most important result of this paper is related to the following two questions in the Möbius gyrovector space $(\mathbb{D}, \oplus, \odot)$.

1. $\{r_1 \odot a_1 \oplus r_2 \odot a_2 \mid r_1, r_2 \in \mathbb{R}\} = \{s_2 \odot a_2 \oplus s_1 \odot a_1 \mid s_1, s_2 \in \mathbb{R}\}$?

2. $r \odot (s_1 \odot a_1 \oplus s_2 \odot a_2) \in \{r_1 \odot a_1 \oplus r_2 \odot a_2 \mid r_1, r_2 \in \mathbb{R}\}$?

They gave an affirmative answer to both Questions 1 and 2.

## 3 Concluding remark

In this paper we survey most important result in literature on the Möbius gyrogroup $(\mathbb{D}, \oplus)$ and Möbius gyrovector space $(\mathbb{D}, \oplus, \odot)$. Ferreira and Ren [8] characterized all subgyrogroups of the Möbius gyrogroup $(\mathbb{D}, \oplus)$. We end this paper with the following question:

**Question 3.1.** Is there any characterization of subgyrovector spaces of $(\mathbb{D}, \oplus, \odot)$?

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
