# Peer review of "On Möbius Gyrogroup and Möbius Gyrovector Space"

_SciPost Physics Proceedings, doi:SciPost Phys. Proc. 14, 041 (2023)_

## Round 1 · Referee Report · Anonymous (Referee 1) · 2023-1-3

Editor
SciPost Physics Proceedings
SciPost Editorial Admin `<submissions@scipost.org>`
December 22, 2022

Referee's Report about
Title: On Möbius Gyrogroup and Möbius Gyrovector Space
by
Kurosh Mavaddat Nezhaad and Ali RezaAshrafi

Gyrogroups and gyrovector spaces form a natural generalization of groups and vector spaces, modeled on Einstein velocity addition law, in which associativity is generalized to *gyroassociativity* and commutativity is generalized to *gyrocommutativity*. The definition of gyrogroups and gyrovector spaces is presented in the paper under review. Hence, in general, the publication of a survey paper about gyrogroups and gyrovector spaces in the Proceedings is a good idea.

However, I cannot recommend the paper under review for publication for the following reasons:

(1) The paper contains several typographical errors and linguistic errors (which can easily be corrected).

(2) Citing from Ref. [6], on the $2^{nd}$ paragraph of p. 2 it is stated that "$u, v, w \in \mathbb{D}$". But, the involved inner products like $u{\cdot}v$ are undefined in $\mathbb{D}$. Rather, they are defined in the unit ball $\mathbb{V}_{s=1}$ of a vector space $\mathbb{V}$, as stated in Ref. [6]. Accordingly, the citation from Ref[6] is erroneous.

(3) On the $4^{th}$ paragraph of p. 2 it is written: "Kinyon and Ungar [8]". But, [8] does not refer to Kinyon and Ungar.

(4) On the last sentence of the $5^{th}$ paragraph of p. 2 it is written: "as tends to infinity". Unfortunately, the object that "tends to infinity" (which is the speed of light in empty space) is not disclosed. In fact, the missing object that "tends to infinity", cannot be disclosed in the paper since the paper implicitly assumes that the speed of light is $c = 1$.

I, therefore, recommend against the publication of the paper in its current form.

---

## Round 2 · Author Response

Following the review, all typos and corrections are considered, corrected, and I read the whole again.
With kind regards,
Kurosh Mavaddat Nezhaad

---

## Editorial Decision

published